# Intralesional Immunotherapy for Non-Genital Viral Warts: A Review of Current Evidence and Future Perspectives

**DOI:** 10.3390/ijms26125644

**Published:** 2025-06-12

**Authors:** Emilia Kucharczyk, Karolina Pawłuszkiewicz, Karol Biliński, Joanna Maj, Małgorzata Ponikowska

**Affiliations:** 1Faculty of Medicine, Wroclaw Medical University, Wybrzeże L. Pasteura 1, 50-367 Wroclaw, Poland; emilia.kucharczyk@student.umw.edu.pl (E.K.); karolina.pawluszkiewicz@student.umw.edu.pl (K.P.); karol.bilinski@student.umw.edu.pl (K.B.); 2Clinical Department of General Dermatology, Centre of General Dermatology Oncodermatology, Wroclaw Medical University, 50-556 Wroclaw, Poland; joanna.maj@umw.edu.pl

**Keywords:** immunomodulation, intralesional therapy, viral warts

## Abstract

Cutaneous warts caused by human papillomavirus (HPV) are among the most common dermatological conditions, affecting the quality of life of numerous people. Although they are widespread, effective and reliable treatment alternatives are limited, emphasizing the necessity for novel treatment options. Intralesional immunotherapy has emerged as a promising alternative, aiming to stimulate the host immune response to achieve the clearance of both treated and distant lesions. This review explores the immunopathogenesis of cutaneous warts and provides an in-depth analysis of intralesional therapies including measles–mumps–rubella (MMR) vaccine, purified protein derivative (PPD), Bacillus Calmette–Guérin (BCG), *Candida* antigen, *Mycobacterium w* vaccine (MWV), vitamin D3, and autoinoculation. We provide a comprehensive analysis of the most promising modalities, highlighting their mechanism of action, outcomes, advantages, and limitations. Although initial data indicate that intralesional immunotherapy offers advantageous efficacy and tolerability, there is a lack of standardized treatment protocols and randomized controlled trials to endorse its broad application. Nevertheless, considering its potential to address local and distant lesions with minimal adverse effects, intralesional immunotherapy may represent a transformative approach to managing cutaneous warts.

## 1. Introduction

Warts are benign epidermal proliferations resulting from infection with the human papillomavirus (HPV), a DNA virus that induces hyperplasia of the epithelium [1]. Over 180 types of HPV have been identified and classified into three categories based on tissue tropism and clinical manifestation: cutaneous (non-genital), mucocutaneous, and those associated with epidermodysplasia verruciformis [2,3]. Non-genital warts can present with a variety of clinical manifestations depending on the HPV type and anatomical location. Common warts, or verruca vulgaris, predominantly occur on the exposed sites of hands and are present as skin-colored, dome-shaped, hyperkeratotic papules with a rough surface. They are associated most with HPV type 2; however, other types have also been reported [4,5,6]. Flat warts, or verruca plana, are elevated papules in color from skin-toned to light brown. They are typically small and mostly seen on dorsal hands and legs. These lesions are primarily caused by HPV types 3, 10, and 28. Plantar warts, or verruca plantaris, are usually firm papules localized deep in the skin. These lesions are typically embedded with a hyperkeratotic layer impacting patients’ walking and daily activities. They are most caused by HPV types 1 and 2 and are highly transmissible through direct contact, particularly in a moist environment [4,7].

The prevalence of non-genital warts varies significantly across age groups, populations, and lifestyle factors. The highest is noted among the pediatric population. In a cross-sectional study conducted in the Netherlands, approximately one-third of examined children aged from 4 to 12 years were found to have cutaneous warts [8]. Similarly, data from Australia indicate a range from 12% in children of 4–6 years old to 24% in 16–18-year-olds [9]. In contrast, the number of studies among the adult population is limited and shows considerable variability across existing studies [5,10].

Although a wide range of topical treatment modalities is available, only salicylic acid and cryotherapy have consistently demonstrated a modest therapeutic effect in the management of cutaneous warts [11]. However, resistant warts remain a clinical challenge, and there are no universal treatment procedures for difficult-to-treat lesions [12,13]. In recent years, intralesional immunotherapy has been reported to be more effective for managing recalcitrant warts than traditional methods [14]. The main mechanism of this promising method is to deliver an antigen directly to the lesion that elicits and modulates an immune response to eliminate the virus [13]. Recent promising reports may broaden therapeutic options for non-genital warts, creating opportunities for optimized therapy not only for resistant lesions but also as a potential first-line therapy.

## 2. Immunological Basis of Anti-Viral Skin Response

The skin functions as the largest organ and a critical component of the immune system, contributing significantly to antiviral defence through the activation of innate immunity and intracellular antiviral pathways. Epidermal cells, such as keratinocytes, detect viruses using pattern recognition receptors (PRRs), including Toll-like receptors (TLRs), and DNA sensors, which trigger the production of type I interferons (IFN) and proinflammatory cytokines [15,16]. The skin not only acts as a physical barrier but is an active immune environment, capable of differentiating responses depending on the type of virus (e.g. herpes simplex virus, HPV, arboviruses) and demographic factors such as age and microbiome [17]. The whole creates a three-level response: innate, adaptive, and intrinsic, providing effective antiviral protection for the skin.

### 2.1. Immune Responses for HPV Clearance

The clearance of HPV involves a coordinated interplay between the innate and adaptive immune responses, as illustrated in Figure 1. Following viral entry into basal keratinocytes through microabrasions, HPV evades early detection by replicating slowly and producing minimal antigens. Despite this, innate immune cells such as natural killer (NK) and natural killer T (NKT) cells play a key role in recognizing infected cells through decreased major histocompatibility complex class I (MHC I) expression and CD1d-presented antigens, respectively [18]. NK cells, upon activation by cytokines such as interleukin (IL)12, IL-15, IL-18, and IFN-α/β, contribute to the antiviral response by producing interferon-gamma (IFN-γ), IL-12, IL-2, and CCL5 and by enhancing cytotoxicity through Killer-cell Immunoglobulin-like Receptor (KIR)-mediated signalling pathways [18,19]. NKT cells, through their interaction with CD1d-presented glycolipids, contribute to viral clearance by releasing granzyme B, perforin, IFN-γ, tumour necrosis factor alpha (TNF-α), and a diverse array of ILs, including IL-2, IL-4, IL-13, and IL-21 [20]. These cytokines augment innate cytotoxic functions and modulate the adaptive immune response; for instance, IL-21 enhances the cytolytic activity of cytotoxic T lymphocytes (CTLs) [21].

Keratinocytes, the primary cellular target of HPV, express a wide range of pattern recognition receptors (PRRs) capable of detecting pathogen-associated molecular patterns (PAMPs) and damage-associated molecular patterns (DAMPs) [15]. These PRRs include TLRs, cytosolic DNA sensors, and others. When activated, these receptors typically trigger immune responses by initiating IFN signalling and promote the release of inflammatory cytokines. PRRs, including TLR9 and the cyclic GMP-AMP synthase—a stimulator of the IFN gene (cGAS-STING) pathway—detect viral DNA within infected cells, leading to the production of type I IFNs and IFN-stimulated genes (ISGs), establishing an antiviral state [22,23]. These innate signals help prime the adaptive immune response. 

Infected keratinocytes and dendritic cells (DCs) produce pro-inflammatory cytokines such as IL-1β, IL-6, and TNF-α, along with chemokines like CXCL10 and CCL20, promoting the recruitment of antigen-presenting cells (APCs) such as Langerhans cells [24,25]. APCs capture viral antigens and migrate to regional lymph nodes, where they present them to CD4^+^ and CD8^+^ T cells. ILs play a vital role: IL-2 promotes T cell proliferation, IL-21 enhances CTL responses, and IL-12 facilitates T helper 1 (Th1) polarization, critical for effective antiviral defence [21]. Activated CTLs migrate back to the site of infection, where they eliminate HPV-infected keratinocytes using perforin, granzymes, and Fas/FasL-mediated pathways. The resulting cellular debris is then engulfed and processed by phagocytes. Meanwhile, helper CD4^+^ T cells support the activation of CTLs, dendritic cell maturation, and B cell antibody production [26]. Tissue-resident memory T cells (TRMs) persist in the epithelium post-clearance, maintaining immune surveillance and enabling rapid responses upon reinfection [27].

### 2.2. Immune Evasion Strategies of HPV

HPV exploits the specificity of the skin’s immune environment to evade early detection and elimination by the immune system effectively. Unlike cytolytic viruses that induce substantial cellular damage and trigger strong inflammatory responses, HPV replicates at very low levels during the early stages of infection, producing minimal viral antigens [28]. In the basal layers of the epithelium, HPV maintains a low copy number and replicates slowly, further reducing its visibility to immune surveillance [29]. As the infection progresses and viral replication increases, the virus ascends into the upper layers of the epidermis. At this stage, both innate and adaptive immune responses are increasingly activated, playing a key role in recognizing and controlling viral spread (propagation/transmission) [17,28].

HPV evades immune detection through a range of mechanisms, including the suppression of IFN responses, the down-regulation of MHC class I and II molecules, and the inhibition of inflammation [30]. Additionally, HPV alters antigen processing machinery, preventing effective T cell epitope generation and hindering the detection of infected cells [16,28]. These strategies protect the virus from immune-mediated cytolysis and apoptosis, allowing for persistent infection. As a result, HPV contributes to the formation of warts and can lead to the progression of malignancies.

As previously noted, keratinocytes—the primary cellular targets of HPV—express PRRs, including TLRs and cytosolic DNA sensors, which detect PAMPs and DAMPs [15]. The activation of these receptors induces IFN signaling and pro-inflammatory cytokine production, thereby initiating the innate immune response. However, HPV evades these immune responses by interfering with these pathways, blocking the release of cytokines and preventing the activation of antigen-presenting cells. This allows HPV to avoid detection and persist within the host [29,30].

Previous studies have shown that a high expression of TLR3, TLR7, TLR8, and TLR9, which recognize viral nucleic acids as part of the innate immune response, is associated with the elimination of HPV [31]. At the same time, HPV reduces the expression of DNA sensors and IFN regulatory factors, impairing the production of IFNs [30,32].

Evidence indicates that HPV proteins, particularly E6 and E7, interfere with IFN production by targeting upstream components of DNA-sensing pathways. E6 interferes with the function of IFN regulatory factor 3 (IRF-3), a critical transcription factor for the induction of IFN-β, by promoting its degradation [32]. This action hampers the production of type I IFNs, weakening the innate antiviral response. Additionally, E7 disrupts the activity of IRF-1 and the IFN-stimulated gene factor 3 (ISGF3) complex, further impairing the IFN signalling pathway [23]. IFNs play a crucial role in clearing the HPV infection. Type I (IFN-α, IFN-β) and III IFNs act as the first responders, creating an antiviral environment and recruiting immune cells, while IFN-γ amplifies adaptive immunity [23]. HPV disrupts all three IFN pathways through viral proteins like E5, E6, and E7, enabling it to persist in keratinocytes and cause lesions like warts [33].

The E5 protein adds another layer of defence by downregulating MHC class I expression and impairing antigen processing in keratinocytes, thus evading CD8+ T cell recognition while sparing HLA molecules that inhibit NK cells. E5 also impairs the immunoproteasome and STING pathway, limiting peptide generation and IFN production [33]. Collectively, these immune evasion strategies enable HPV to bypass the skin’s antiviral defences, persist within keratinocytes, and drive the visible proliferation of infected cells, resulting in wart formation [32,33]. Table 1 presents the sources, immune functions, and HPV-mediated evasion strategies related to IFN-driven antiviral responses [23,32,33].

The immune-based approach presented in Table 1 is further supported by clinical observations. A 51-year-old female patient with a 5-year history of treatment-resistant periungual warts, unresponsive to multiple topical, intralesional, and systemic therapies, exhibited a marked clinical response to systemic IFN alpha-2b. This improvement was accompanied by restoration of NK cell cytotoxicity, suggesting that IFN-α may exert its therapeutic effect by correcting underlying NK cell dysfunction. The case underscores the potential of IFN-α as a targeted immunomodulatory strategy in non-genital, recalcitrant HPV-related warts, particularly in patients with subtle immune dysregulation [34].

In addition to its observed benefit in patients with subtle NK cell dysfunction, IFN-α-2b demonstrated marked clinical efficacy in a patient with Dedicator of Cytokinesis 8 (DOCK8) deficiency, a primary immunodeficiency associated with severe, recalcitrant cutaneous warts. The therapeutic response correlated with profound IFN-α deficiency due to a paucity of circulating plasmacytoid dendritic cells (pDCs) and impaired TLR-mediated signalling. IFN-α supplementation likely compensated for this innate immune defect by restoring NK and cytotoxic T cell function, as well as MyD88-dependent antiviral pathways. While hematopoietic stem cell transplantation remains the definitive treatment for DOCK8 deficiency, IFN-α may serve as a valuable adjunct or bridging therapy in controlling HPV-driven disease in such patients [23]. DOCK8 deficiency not only impacts innate immunity but also terminates adaptive immune responses. It disrupts T helper (Th) cell differentiation by reducing Th1 and Th17 cells but increasing Th2 responses, impairing antiviral immunity. It also impacts B cell receptor (BCR) signalling, impairing germinal centre (GC) formation and normal antibody production—both crucial components of humoral immunity necessary to control HPV infections [35].

## 3. Intralesional Immunotherapy

Intralesional immunotherapy has emerged as a promising alternative for the management of non-genital viral warts, owing to its potential to induce not only local but also distant wart resolution through systemic immune activation. A variety of agents—such as the Measles–Mumps–Rubella vaccine (MMR), Purified Protein Derivative (PPD), the Bacillus Calmette–Guérin vaccine (BCG), the *Mycobacterium w* vaccine (MWV), autoinoculation, *Candida albicans* antigen, *Trichophyton* antigen, IFN-α, and vitamin D3—have been utilized with varying success [36].

### 3.1. Immunological Insights into Intralesional Therapy for HPV-Induced Wart

Since HPV can suppress local immune responses, effective treatment for non-genital warts often requires therapeutic strategies that reactivate the host’s immune system. Intralesional immunotherapy—a technique involving the injection of immunostimulatory agents directly into wart tissue—has gained considerable support in recent studies as a potent method for inducing an antiviral immune response [15,16]. Various intralesional agents—including MMR vaccination, PPD, BCG, MWV, and autoinoculation—exert their therapeutic effects primarily through a type IV delayed-type hypersensitivity [37,38,39,40]. The mechanism of delayed-type IVa immune response in HPV wart clearance is presented in Figure 2 [37,41].

The immunological mechanisms underlying the activity of IFN-γ, *Candida*, and *Trichophyton* antigens, as well as vitamin D, are comprehensively delineated within their respective subsections.

### 3.2. Measles-Mumps-Rubella (MMR) Vaccine

#### 3.2.1. Characterization, Therapeutic Potential, and Clinical Considerations of the Investigated Agent

The MMR vaccine contains live attenuated viruses of three pathogens: measles virus, mumps virus, and rubella virus. The MMR vaccine is believed to act through a nonspecific inflammatory response to viral antigens, which enhances the host’s immune reactivity to HPV. While the exact immunological mechanism is not fully understood, the combination of measles, mumps, and rubella antigens appears to promote a Th1-dominant response and a type IV delayed-type hypersensitivity [42,43].

The MMR vaccine offers several therapeutic advantages, including the resolution of both injected and distant, untreated warts, with a minimal risk of scarring, low recurrence rates, and a favourable safety profile [6,43]. However, its use must be approached with caution in immunocompromised patients, who, while more susceptible to persistent HPV infections, are also at increased risk of adverse effects from live-attenuated vaccines. Common drug reactions associated with MMR-based immunotherapy are generally mild and self-limiting, including local pain, pruritus, erythema, swelling at the injection site, and transient flu-like symptoms [37,44]. Overall, the MMR vaccine represents a safe, effective, and well-tolerated therapeutic modality for the treatment of cutaneous warts [44].

Intralesional MMR vaccine immunotherapy has shown promise in the treatment of paediatric warts, particularly in cases presenting multiple lesions [45]. In a retrospective study by Na et al., patients received injections into the largest wart every two weeks, contingent on a positive test dose, following Johnson et al. [46,47]. A clinical response in the primary lesion was observed in 51.5% of cases, while 46.7% of patients with distant warts exhibited partial or complete resolution. Notably, younger patients responded more favourably, and no treatment-limiting adverse effects were reported [47]. Limited information is available on the use of the MMR vaccine for the treatment of facial warts. Al-Qassabi AM et al. in their study report two cases of resistant facial warts treated with a single intralesional injection of the MMR vaccine, resulting in complete resolution [48]. Patient 1 received 0.2 mL MMR vaccine into the two largest warts, while Patient 2 received 0.1 mL bilaterally into the largest papule, with both showing successful outcomes without complications, except mild temporary pain during injection [48]. Despite encouraging results, further high-quality studies are warranted to conclusively establish the efficacy and safety of intralesional MMR immunotherapy in the treatment of facial warts. 

In a case report by Al-Qassabi and Al-Farsi, two Omani male patients with recalcitrant genital warts achieved complete resolution following intralesional treatment with the MMR vaccine, highlighting the potential of this immunotherapeutic approach—also explored in non-genital wart management—for resistant HPV-related genital lesions [49].

#### 3.2.2. Dosage and Administration

Due to the absence of a standardized protocol for the minimum effective dose and optimal treatment duration, most studies administer three to six intralesional MMR vaccine doses ranging from 0.1 to 0.5 mL, administered at 2–3-week intervals [50]. Table 2 presents a summary of published protocols for intralesional MMR vaccine administration in the management of non-genital viral warts [50,51,52].

Pre-sensitization has been shown to enhance the immunogenicity and therapeutic efficacy of intralesional MMR vaccine administration in the treatment of non-genital viral warts. Awal G and Kaur S demonstrated that the intradermal administration of 0.1 mL of the MMR vaccine to the volar forearm, followed by evaluation for local erythema or induration after one week, can effectively identify immunological responsiveness [53]. In cases with suboptimal response, the escalation of either vaccine volume or treatment frequency has been proposed as a strategy to enhance therapeutic outcomes.

While many studies have reported follow-up durations ranging from 8 to 24 weeks post-treatment, a few, such as Sobhy MN et al. [54], extended follow-up to six months, demonstrating an 86.9% complete response rate, with no response in the control group. In contrast, Rageh RM et al. [55] observed a 26.7% clearance rate with 53.3% of cases showing no response in their shorter follow-up study. Kaur A et al. [50] further highlighted prolonged efficacy, showing continued improvement up to 24 weeks post-treatment in the MMR group, while the MIP group showed no further change after 16 weeks. Varghese A et al. also demonstrated a delayed therapeutic response—complete clearance was observed after an extended follow-up, which occurred 13 weeks post-treatment [44]. These findings collectively underscore the importance of extended follow-up in immunotherapy studies for cutaneous warts, as delayed or progressive responses may not be fully captured by shorter observation periods.

### 3.3. Purified Protein Derivative (PPD), the Bacille Calmette–Guérin (BCG) VAccine and the Mycobacterium W Vaccine (MWV)

#### 3.3.1. Characterization, Therapeutic Potential, and Clinical Considerations of the Investigated Agent

Other antigens employed in intralesional immunotherapy are also derived from *Mycobacterium tuberculosis*, the causative agent of tuberculosis. Three distinct formulations are commonly utilized: PPD, MWV vaccine, BCG vaccine [37]. 

PPD is a non-infectious extract derived from *Mycobacterium tuberculosis*, primarily used as a diagnostic tool for tuberculosis through intradermal injection and subsequent delayed-type hypersensitivity reaction. It contains denatured proteins from tubercle bacilli and, unlike live vaccines, does not pose a risk of active infection [56]. PPD, when injected into wart tissue, activates memory T cells sensitized through prior exposure to mycobacterial antigens (e.g., via BCG vaccination or environmental contact) and exerts its therapeutic effects primarily through a type IV delayed-type hypersensitivity. The cytokine changes observed after PPD administration are summarized in Table 3 [57,58,59,60]. 

Cytokine profiling across studies reveals a complex and variable immunological response to PPD immunotherapy. The consistent reduction in IL-10 observed across studies underscores its suppressive role in wart persistence through the inhibition of Th1 cells, cytotoxic T lymphocytes, and antigen-presenting cells, whereas IL-1 upregulation may serve as a key indicator of therapeutic efficacy specific to PPD [60].

BCG is a live, attenuated strain of *Mycobacterium bovis* and has long been utilized as a vaccine against tuberculosis in numerous countries. Beyond its established role in tuberculosis prevention, BCG has also shown therapeutic potential in dermatology. Several case reports and clinical observations have documented its successful application in managing recalcitrant extragenital warts, with complete resolution often observed after three to five intralesional injections [61]. In a study by Sharquie et al., BCG immunotherapy for viral warts was well-tolerated, with no significant local or systemic adverse events. Interestingly, certain inflammatory signs were commonly observed during wart regression, including pruritus in all patients, transient enlargement of lesions in 80%, and localized tenderness in 30%. Given its low cost and favourable safety profile, BCG may represent a practical and accessible treatment option, particularly in low-resource settings [62].

The Mw vaccine, recently approved for use in leprosy patients, consists of *Mycobacterium indicus pranii (MIP)*—formerly known as *Mycobacterium w*—a nonpathogenic, cultivable, atypical mycobacterium classified under Runyon group IV. This heat-killed suspension has long been utilized in the management of various immune-mediated and infectious diseases, demonstrating its multipurpose immunomodulatory potential [63,64].

In the clinical study by Sil et al., these immunological changes were confirmed: IFN-γ and IL-1 levels increased significantly following Mw vaccine therapy, while immunosuppressive cytokines such as IL-10 were downregulated. The observed reduction in IL-10 was strongly associated with lesion regression, underscoring its role in viral persistence and the suppression of host immune response [60].

Reported adverse effects of intralesional immunotherapy are generally mild and self-limiting. These may include localized reactions such as pain, nodularity, sterile pustule formation, ulceration, and occasional scarring at the injection site. Systemic symptoms such as transient fever, flu-like manifestations, and regional lymphadenopathy have also been observed. Additionally, some patients may experience paraesthesia in the limb distal to the site of administration, though such neurological symptoms are infrequent [65,66].

#### 3.3.2. Dosage and Administration

A literature review of comparative studies, multicentre trials, and randomized controlled trials (RCTs) published between 2020 and 2025 (for PPD and BCG) and between 2016 and 2025 (for Mw vaccine) identified key clinical data on intralesional immunotherapy for viral warts. Table 4 summarizes treatment protocols, cytokine responses, and reported clearance rates [67,68,69,70,71,72,73].

### 3.4. Autoinoculation Techniques

#### 3.4.1. Characterization, Therapeutic Potential, and Clinical Considerations of the Investigated Agent

Autoinoculation is an emerging immunotherapeutic method for the management of multiple and treatment-resistant viral warts. Figure 3 illustrates the protocol for autoinoculation employed in the therapeutic management of non-genital viral warts [74,75,76].

In a modified technique, Nischal et al. employed excised wart tissue for autoinoculation into the subcutaneous tissue [77].

Adverse events associated with autoinoculation are generally minimal and include pustule formation, hypertrophic scarring at the injection site, and post-inflammatory changes such as hypopigmentation or hyperpigmentation. In the study by Ashraf et al., two (2.1%) patients developed infection at the autoinoculation site. A higher incidence of infection (13.33%) was reported in a study by Taneja et al [78]. However, these infections did not significantly affect the treatment outcomes [78]. The primary limitation of this study was the small sample size, highlighting the need for future research with larger cohorts. Additionally, as this was a single-centre study conducted in an urban setting, the results may not be generalizable to broader populations [40].

#### 3.4.2. Dosage and Administration

A focused literature search on PubMed using the terms (autoinoculation) AND (warts) and filtering for studies published between 2020 and 2025 identified a set of clinical trials and observational studies investigating the role of autoinoculation in the treatment of non-genital viral warts. Table 5 summarizes the key characteristics and outcomes of relevant studies, including the number of autoinoculation sessions, patient monitoring timelines, and observed efficacy rates [39,40,78,79].

Autoinoculation of non-genital warts has reported optimistic clearance rates—up to 91.66% in some studies [40]—with minimal side effects such as infection or pigmentation change. Results, although promising, must be validated in larger and more diverse populations [39,40,78,79].

### 3.5. Candida Antigen 

#### 3.5.1. Mechanism of Action, Advantages, and Limitations of the Applied Therapeutic Approach

*Candida* antigen has been documented as an effective treatment modality for warts since 1979 [2]. Nakagawa et al. demonstrated that T-cell proliferation was markedly enhanced upon stimulation with *Candida*-pulsed Langerhans cells compared to unstimulated controls (*p* = 0.004), with IL-12p40 identified as the most abundantly expressed cytokine at both mRNA and protein levels. Moreover, *Candida* stimulation induced the transcription of multiple cytokine genes, including IL-23Ap19, IFN-γ, IL-1β, IL-2, IL-6, IL-8, and IL-10. Their findings further indicated that Langerhans cells express a broad range of PRRs and that dectin-1 blockade suppressed IL-12p40 production in certain donors. These results suggest that *Candida* facilitates Th1 polarization among others by the dectin-1-mediated activation of Langerhans cells [42]. 

Hammad et al. investigated the immunological basis of clinical responses to *Candida* antigen immunotherapy in 44 patients with cutaneous warts, revealing that although more than half of the patients achieved complete clearance (56.4%), serum levels of C3c and TNF-α were significantly elevated among non-responders and showed a strong correlation, suggesting that heightened complement pathway activation and a robust Th1 cytokine response may contribute to treatment resistance [80].

Cutaneous warts are frequently observed after kidney transplantation, with plantar warts being the most common manifestation [81]. Although intralesional *Candida* immunotherapy has demonstrated promising efficacy in treating recalcitrant warts in both immunocompetent and immunocompromised patients [82,83,84], its safety profile in transplant recipients remains uncertain, mainly due to concerns regarding excessive immune activation potentially leading to allograft rejection [85]. Acharya et al. observed a temporal association between IL *Candida* immunotherapy and the subsequent development of BK viremia (i.e. the presence of BK polyomavirus DNA in the bloodstream), elevated donor-derived cell-free DNA (dd-cfDNA), the emergence of donor-specific antibodies (DSA), and *Pneumocystis jirovecii* pneumonia. Dd-cfDNA, a non-invasive biomarker, reflects DNA release from injured graft cells and is increasingly used to detect subclinical allograft injury or early rejection before changes in renal function [81,86]. DSAs, directed against donor-specific human leukocyte antigens, are strongly associated with antibody-mediated rejection and poorer graft outcomes. The observation of elevated dd-cfDNA and DSA following intralesional *Candida* therapy raises concerns about systemic immune activation extending beyond localized wart treatment. However, no direct causal relationship between intralesional Candida immunotherapy and these findings could be established [81,86]. Further studies are urgently required to elucidate the safety profile of intralesional *Candida* immunotherapy in paediatric kidney transplant recipients and to optimize post-treatment monitoring strategies [86,87].

Adverse effects associated with *Candida* immunotherapy include febrile reactions, myalgia, injection site pain, erythema, oedema, and, rarely, the painful purple digit syndrome. Nevertheless, this therapeutic approach remains highly cost-effective [56].

#### 3.5.2. Dosage and Administration

The literature search was conducted using the terms (*Candida* antigen) AND (Warts), limited to studies published between 2020 and 2025. Table 6 summarizes key findings from these clinical studies, detailing the type of warts treated, the administered dose of *Candida* antigen per session, injection protocols, and reported clearance rates [55,67,88,89,90,91,92,93,94].

### 3.6. Trichophyton Antigen

#### 3.6.1. Mechanism of Action, Advantages, and Limitations of the Applied Therapeutic Approach

Species of *Trichophyton*, a genus of keratinophilic fungi, exhibit a predilection for colonizing keratinized human structures—the stratum corneum of the skin, nails, and hair shafts. *Trichophyton tonsurans*, *T. mentagrophytes*, and *T. rubrum* are recognized as the principal aetiological agents responsible for the majority of superficial cutaneous mycoses worldwide, including tinea capitis, tinea pedis, and onychomycosis. The incidence of symptomatic disease increases with age, while asymptomatic colonization is remarkably common in the adult population, with estimated carriage rates ranging from 30% to 70% [95,96]. *Trichophyton* antigen, prepared from *Trichophyton* species via allergenic extraction, has demonstrated immunotherapeutic potential. In a randomized, single-blind trial, Horn et al. reported a 62% response rate in patients with cutaneous warts treated intralesionally [97]. 

#### 3.6.2. Dosage and Administration

The treatment involves the intralesional injection of *Trichophyton* antigen into the largest wart, with a dosage of 0.3 mL per injection. This procedure is typically repeated every three weeks, with a maximum of five sessions [44].

### 3.7. Interferons α-2B (IFNα-2B)

#### 3.7.1. Mechanism of Action, Advantages, and Limitations of the Applied Therapeutic Approach

IFNs are naturally occurring small proteins with crucial in vivo antiviral functions, including viral interference. Among them, IFNα-2B has demonstrated broad clinical utility due to its potent immunomodulatory, antiviral, and antiproliferative properties [98]. A randomized, single-blind trial by Horn et al., assessed 233 individuals presenting with one or more cutaneous warts. The study found no significant therapeutic advantage from the addition of IFNα2a or IFNα2b to common immunotherapeutic agents such as *Candida*, *Trichophyton*, or mumps antigens [97]. 

Adverse effects commonly associated with IFN therapy encompass a constellation of flu-like symptoms, including pyrexia, chills, fatigue, headache, myalgia, nausea, vomiting, and diarrhoea. The intensity and frequency of these reactions appear to be dose-dependent [97,98]. The administration of a low-dose regimen of IFNα resulted in transient flu-like symptoms in all patients receiving active treatment in the Varnavides et al. study [99]. Similarly, Lee et al. reported systemic symptoms in 71% of subjects receiving high-dose IFNα and 26% in the low-dose group [100]. In contrast, Vance et al. documented study withdrawal in five participants out of 100 within the high-dose cohort—two due to localized injection site pain, and three owing to systemic flu-like reactions [101]. These findings underscore the need for more comprehensive safety studies to fully characterize the tolerability profile of IFNα across varying dosing strategies.

#### 3.7.2. Dosage and Administration

The treatment involves the intralesional injection of IFN alpha into each lesion, with a dosage of 1–2 million units. This procedure is typically repeated every three days for 3 weeks [66].

### 3.8. Vitamin D3

#### 3.8.1. Mechanism of Action, Advantages, and Limitations of the Applied Therapeutic Approach

A fat-soluble vitamin D3 exerts its biological effects through endocrine, autocrine, and paracrine pathways. Its endocrine role primarily involves the regulation of serum calcium homeostasis. In contrast, its autocrine and paracrine functions are mediated by its interaction with nuclear vitamin D receptors (VDRs) expressed in various cell types, enabling localized immunological and cellular regulatory actions [102].

The precise mechanism by which intralesional vitamin D3 treats cutaneous warts is not fully elucidated. It is proposed that vitamin D3 enhances innate immunity by stimulating the production of antimicrobial peptides such as defensin β2 and cathelicidin in keratinocytes, monocytes, and macrophages, thereby augmenting their antimicrobial, chemotactic, and phagocytic functions [103]. Some studies have shown minimal or no increase in IL-12 following vitamin D3 treatment compared to other immunotherapeutics like PPD. IL-12 is important for early Th1 polarization, but vitamin D3 seems to primarily act downstream via direct IFN-γ induction [59]. Moreover, intralesional vitamin D3 may exert its effects via the upregulation of VDR and hydroxylase genes, stimulating local immune responses [104]. 

Common side effects associated with intralesional vitamin D3 therapy include localized itching, swelling, pain at the injection site, and dyspigmentation [92,102]. In addition, Nasr et al. reported the occurrence of a vasovagal reaction following injection, and no cases of post-treatment fever were observed [92].

#### 3.8.2. Dosage and Administration 

The literature search was performed using the terms (Vitamin D3) AND (Warts) limited to studies published between 2020 and 2025. Table 7 summarizes key findings from these clinical studies, highlighting the type of warts treated, the administered dose of vitamin D3 per session, injection schedules, and reported clearance rates [52,92,102,103,105,106,107,108].

## 4. Conventional Intralesional Therapy

### 4.1. Bleomycin 

Bleomycin is an antineoplastic antibiotic that exerts its therapeutic effects on warts by inducing DNA strand breaks, which lead to the apoptosis of infected keratinocytes [109]. Intralesionally administered bleomycin causes local cytotoxicity. This direct cytotoxic effect is complemented by the induction of a local inflammatory response, which may additionally aid in the clearance of HPV-infected cells. A 2024 systematic review by Mullen et al. analyzed 12 randomized controlled trials involving 927 patients treated with intralesional bleomycin for cutaneous warts. The review reported a median complete clearance rate of 84%, establishing bleomycin as one of the most effective intralesional agents [110]. Despite its high efficacy, the review noted a relatively higher incidence of localized pain and inflammatory reactions compared to other treatments. Additionally, a 2022 analysis by Martin et al. evaluated 26 studies on plantar wart treatments across six intralesional therapies–bleomycin demonstrated a median cure rate of 74% [111]. This therapy can also cause potential local skin reactions like erythema, oedema, and, rarely, nail dystrophy or Raynaud’s phenomenon. Systemic toxicity is rare due to the low doses used [112]. 

### 4.2. Fluorouracil (5-FU) 

5-Fluorouracil (5-FU) is a pyrimidine analogue that inhibits thymidylate synthase, disrupting DNA synthesis and inducing apoptosis in rapidly dividing cells, such as those infected by HPV [113]. When administered intralesionally, 5-FU targets the hyperproliferative keratinocytes within warts, leading to lesion resolution. 

Recent studies have demonstrated the efficacy of intralesional 5-FU in treating recalcitrant non-genital warts. In a randomized controlled trial, Yazdanfar et al. reported a complete response rate of 64.7% in warts treated with a combination of 5-FU, lidocaine, and epinephrine, compared to 35.3% in the placebo group [113]. The treatment was well-tolerated, with no significant systemic adverse events or differences in recurrence rates between groups. Another study by Sepaskhah et al. compared intralesional 5-FU with cryotherapy in patients with common and palmoplantar warts. The complete response rate was 64.3% in the 5-FU group versus 35.7% in the cryotherapy group. Although the difference was not statistically significant, patients receiving 5-FU experienced less pain and greater improvement in quality of life [114]. A systematic review by Martin et al. reported a median cure rate of 59% for intralesional 5-FU across various studies, while bleomycin demonstrated a median cure rate of 74% [108]. A comparative study evaluating 5-FU, *Candida albicans* antigen, and bleomycin reported that 5-FU was the least effective treatment modality, with complete response observed in 45% [91]. In contrast, *Candida* antigen immunotherapy achieved a 60% complete response rate, while bleomycin was the most effective, with 85% complete clearance rate. 

### 4.3. Acyclovir

Acyclovir is a widely utilized antiviral agent for the treatment of infections caused by herpes simplex virus (HSV) and varicella-zoster virus (VZV) [115]. It is an acyclic purine nucleoside analogue that acts as a DNA synthesis and viral replication inhibitor following its conversion to the active metabolite, acyclovir triphosphate, by viral and cellular enzymes. Because of the ability of the viruses to code a viral thymidine kinase necessary for the initial phosphorylation step, acyclovir is only activated in infected cells. This selectivity minimizes cytotoxicity to uninfected host cells [116]. Recently, the antiviral properties of acyclovir have garnered attention for its potential use as an intralesional therapeutic agent in the treatment of cutaneous warts caused by HPV. 

In a 2025 retrospective study involving 14 patients, the efficacy and safety of intralesional acyclovir for the treatment of palmoplantar and ungual warts were evaluated. A 50 mg/mL solution of acyclovir was injected directly into the wart at a dose of 0.1 mL. Following 5 treatment sessions, 82% of warts showed complete response and 18% exhibited partial response. Adverse effects occurred in 23% and included local tenderness, indicating favourable safety profile of the modality [117]. In a comparative study of intralesional acyclovir versus intralesional PPD, 60% complete response was reported, while 30% in intralesional PPD treatment [118]. However, all individuals treated with acyclovir experienced a burning sensation, and 90% reported pain. In contrast, a more favourable safety profile was presented by PPD, where 60% of patients had no adverse effects and only 40% experienced pain. While intralesional acyclovir appears to be an effective treatment modality, attention should be given to the frequency of its adverse effects [118].

### 4.4. Cidofovir

Cidofovir is an antiviral agent originally approved by the Food and Drug Administration (FDA) for the treatment of cytomegaly retinitis in patients with acquired immunodeficiency syndrome (AIDS) [119]. Although it presents high activity against various DNA viruses, the use of cidofovir has been limited due to its high cost and lack of clinical trials [120]. However, with its availability in generic form, it may be beneficial for patients with recalcitrant HPV warts. Cidofovir, as an acyclic nucleoside phosphonate, targets viral DNA polymerase. Its active dephosphorylated metabolite exhibits higher affinity for viral DNA polymerase compared to the polymerase of the host. It acts as a competitive inhibitor of deoxycytidine triphosphate incorporation into viral DNA, thereby limiting viral DNA synthesis [121].

Cidofovir has demonstrated promising efficacy in the treatment of recalcitrant non-genital warts. In a case series conducted at the University of California, intralesional cidofovir was used in nine patients with recalcitrant periungual warts [122]. All patients showed improvement, with seven of them achieving near-complete resolution. Adverse effects were mild and well tolerated, including primarily localized pain, oedema, and blister formation. The efficacy of cidofovir in the treatment of refractory cutaneous verrucae was confirmed by a retrospective study conducted by Anshelevich E. et al. Among the cohort of 58 patients, 98.3% showed clinical improvement, and 75.9% resolution following a mean of 3.4 treatment sessions [123]. Adverse effects were rare and limited to local reactions, including blistering, pain, and swelling. 

### 4.5. Zinc Sulfate 

The oral administration of zinc has previously been reported as a promising and effective option for the treatment of non-genital warts [124]. Zinc is a pivotal element and serves as a cofactor for many enzymes and proteins involved in crucial cellular processes such as division, growth and development [125]. When injected intralesionally, zinc sulphate promotes the migration of eosinophiles, followed by lymphocytes and fibroblasts, to the site of the treated wart [126]. 

A meta-analysis conducted by Martin A. et al., which screened 26 articles, reported a 70% cure rate associated with zinc sulphate [111]. In a trial comparing the efficacy of the treatment of cutaneous warts intralesional *Candida albicans* antigen with 2% zinc sulphate solution, zinc sulphate showed a complete response rate of 68.6%, which was comparable to the 74.3% complete response in a *Canida* antigen group [93]. Another study comparing the intralesional zinc sulphate with vitamin D3 found that zinc was nearly as effective (71.4%) as vitamin D3 (62.9%) [127]. There is a study that reported a higher complete response rate of 80%, although this may be attributed to the inclusion of only common warts [90,128]. While adverse events regarding zinc sulphate did not necessitate the discontinuation of the treatment, it is worth noting that patients receiving zinc sulphate reported severe pain during the injection, whereas individuals injected with vitamin D3 experienced mild discomfort [127]. Contrary to this, Youssef E. et al. noted that treatment with zinc sulphate is less painful than *Candida* antigen and concluded that it is a cheap, effective, and safe procedure that can serve as an alternative modality for all types of non-genital warts, new and resistant [93].

### 4.6. Methotrexate

Intralesional methotrexate (IL-MTX) is a well-established treatment modality for many dermatological conditions, including keratoacanthomas, squamous cell carcinomas, cutaneous lymphomas, and inflammatory skin disease such as psoriasis [127]. Methotrexate (MTX) is an anti-metabolite of folic acid that indirectly inhibits cell division by targeting folate-dependent enzymatic pathways [129]. Precisely, it inhibits dihydrofolate reductase, thereby preventing the reduction of dihydrofolate to tetrahydrofolate. Tetrahydrofolate plays a pivotal role in nucleotide synthesis pathways, essential to the repair, synthesis, and replication of DNA. These properties of methotrexate particularly affect cells with a high mitotic index, such as epithelial cells infected with HPV [130]. 

Studies investigating the use of IL-MTX for the treatment of non-genital warts are limited, making it difficult to assess the efficacy of this approach. In a study conducted by Hamed M. et al., the treatment of non-genital warts with IL-MTX resulted in a complete response rate of only 6.7%. The difference in efficacy between MTX and saline was not statistically significant, suggesting limited therapeutic utility [131]. However, a subsequent study employing full-concentration IL-MTX solution at a concentration of 25 mg/mL administered to the warts of 20 individuals reported more promising outcomes. Complete clearance was observed in seven patients and partial improvement in another 8. Importantly, adverse effects were mild: nine participants reported no side effects, while pain and bruising were reported by eight and six, respectively. Although these findings are more favourable, it is important to note that the sample size was limited to 20 patients [132].

## 5. Comparison of Novel Immunomodulatory Intralesional Techniques

In 2018, Salman et al. conducted a comprehensive meta-analysis of 17 randomized controlled trials comparing intralesional immunotherapy to cryotherapy, placebo, and imiquimod [133]. The authors concluded that PPD and MMR were the most effective procedures in achieving complete recovery. Additionally, intralesional immunotherapy was associated with fewer adverse events compared to standard non-genital warts management. 

Table 8 summarizes recent clinical trials and provides an overview of the latest publications comparing novel immunomodulatory intralesional techniques in managing non-genital warts. This table includes clinical trials published from 2020 to 2025 that compare immunomodulatory intralesional therapies, are indexed in PubMed database, and involve study populations of at least 40 patients [50,55,67,70,92,107].

The data summarized in Table 8 present variability; however, most presented studies report high complete clearance rates with *Canida* antigen, MMR, PPD, and MIP and significantly lower with vitamin D3. Notably, there is a lack of large-population clinical trials, especially randomized controlled trials, investigating intralesional immunotherapy for the treatment of non-genital warts. The latest meta-analysis published in 2024 regarding intralesional immunotherapy also noted high variability across over 500 screened articles and emphasized the need for further research to define efficacy of every treatment [110].

## 6. Future Perspectives and Research Directions

### 6.1. Cytokine Profiles

Emerging evidence suggests that specific cytokines and immune cell markers may serve as predictors of response to intralesional immunotherapy in HPV warts. As mentioned before, Th1 cytokines such as IL-2, IL-12, and IFN-γ are associated with enhanced cell-mediated immunity and have been implicated in wart regression [15,17,21,30]. Conversely, the Th2 cytokine IL-10 may suppress immune responses, potentially correlating with treatment resistance. Analysis of the patient’s immune response by examining the cytokine profile suggests the possibility of early assessment of wart resistance to treatment, which will enable the faster implementation of effective immunotherapy, which, due to its systemic action, will also act preventively against the development of changes in new places [58,60,80,134,135]. 

#### 6.1.1. Interleukin-17A (IL-17A)

Additionally, the presence of regulatory T cells (Tregs), characterized by FoxP3 expression, may modulate immune responses during therapy. The 2021 study by Nassar et al. explored whether serum cytokine levels—specifically IL-17A and macrophage migration inhibitory factor (MIF)—can predict therapeutic response to intralesional *Candida* antigen immunotherapy in patients with multiple common warts (caused by HPV). The results of the study indicate that serum IL-17A and MIF levels are promising immunological biomarkers in the treatment of HPV warts. A decrease in IL-17A and an increase in MIF after *Candida* antigen therapy reflect successful immune reactivation and wart clearance. Importantly, IL-17A levels below 316 pg/mL may predict a favourable clinical outcome, offering a potential tool for personalized immunotherapy strategies for warts [134]. However, studies have shown inconsistent associations between these biomarkers and clinical outcomes, indicating the need for further research to validate their predictive value.

#### 6.1.2. Interleukin-18 (IL-18)

The study by Korsa (2022) supports IL-18 (IL-18) as a promising biomarker of immunotherapeutic response in patients with cutaneous and genital warts treated intralesionally with PPD. Following three sessions of intralesional PPD injections, IL-18 serum levels significantly increased—particularly in clinical responders—indicating that IL-18 may reflect effective immune activation and serve as a predictive indicator for treatment success [135]. 

#### 6.1.3. Interleukin-4 (IL-4) and IFN-γ

A study by Hammad (2021) involving bivalent HPV vaccine stimulation found that an elevated IFN-γ response (≥56.8%) reflects strong cellular immunity and is associated with favourable outcomes in immunotherapy using the bivalent HPV vaccine. In contrast, IL-4 supports Th2-type responses, which are generally less effective against intracellular pathogens like HPV [15,17]. A reduced IL-4 response (≥16.6%) may reflect a shift away from Th2 dominance, reinforcing the IFN-γ-driven Th1 response. Together, these cytokine profiles help predict a patient’s likelihood of therapeutic success [83]. 

However, at the same time both intralesional PPD and MMR vaccines demonstrate comparable efficacy in treating multiple warts, yet they differ subtly in their immunological profiles [58]. IL-12, a Th1 cytokine pivotal for inducing IFN-γ and promoting cell-mediated immunity [17], showed a significant post-treatment increase, particularly with MMR, suggesting a stronger Th1-biased response, which is essential for viral clearance [58]. Conversely, IL-4, a Th2 cytokine known to dampen Th1 responses was unexpectedly increased after PPD treatment, indicating a more mixed Th1/Th2 profile. The elevation of both IL-12 and IL-4 suggests that immunotherapy modulates the immune balance rather than strictly polarizing it, and their levels may reflect treatment-specific immune shifts. Notably, MMR’s greater enhancement of IL-12 supports its stronger activation of antiviral immunity. While an IL-12 increase correlates with effective clearance, the rise in IL-4 after PPD may temper the Th1 response, hinting at a potential limitation or regulatory feedback unique to this antigen [58].

#### 6.1.4. Interleukin-10 (IL-10)

IL-10 is an anti-inflammatory Th2 cytokine that acts by inhibiting the Th1 immune response necessary to clear HPV-infected keratinocytes [28]. In the context of wart immunotherapy, particularly with agents like PPD or MMR vaccines, a significant downregulation of IL-10 correlates with successful therapeutic outcomes. This downregulation reduces immune suppression, thereby enhancing cell-mediated immunity, notably IFN-γ and IL-1 activity. In the cited study, patients whose warts were completely cured had significantly reduced IL-10 levels, especially in the PPD group, suggesting that IL-10 may serve as a negative predictive biomarker [60].

### 6.2. Combination Therapies with Intralesional Immunotherapy

#### 6.2.1. PPD + Cryotherapy

Two studies evaluated the combination of PPD with cryotherapy. While the overall response rates were similar to monotherapy, the combination led to a shorter time to clearance, suggesting a potential benefit in accelerating treatment response [136,137].

#### 6.2.2. *Candida* + Cryotherapy

The study by Attwa et al. (2020) assessed the safety and effectiveness of combining both treatments. Sixty patients were divided into three groups: *Candida* antigen alone, cryotherapy alone, and a combination of both (cryo-immuno-therapy). Over five sessions and a 3-month follow-up, the combination therapy achieved a 40% complete clearance rate—significantly higher than the 20% seen with *Candida* antigen alone. The study concludes that cryo-immuno-therapy is a safe, cost-effective, and more efficacious alternative for treating multiple common warts [138].

#### 6.2.3. Intralesional *Candida* + Oral Isotretinoin

A randomized controlled trial compared the efficacy of intralesional *Candida* antigen, oral isotretinoin, and their combination in treating plane warts. The study found that *Candida* antigen alone had the highest complete clearance rate (55.6%), followed by isotretinoin alone (44.4%) and the combination therapy (38.8%). This suggests that combining these therapies may not provide additional benefits over monotherapy [91].

#### 6.2.4. Intralesional *Candida* + Oral Acitretin

A study by Nofal et al. evaluated the effectiveness of combining acitretin and intralesional *Candida* antigen for treating recalcitrant warts, compared to each agent alone. Among 108 adult patients with long-standing, treatment-resistant warts, the combination therapy achieved complete clearance in 66.7% of cases—significantly higher than acitretin alone (38.9%) or *Candida* antigen alone (33.3%) [139].

#### 6.2.5. PPD + *Candida* Combination Therapy for Plane Warts

A randomized controlled trial by Nofal et al. (2020) found that alternating intralesional injections of PPD (0.1 mL) and *Candida* antigen (0.1 mL, 1/1000 dilution) every 2 weeks for up to six sessions led to significantly higher therapeutic response—both complete and partial—compared to either agent alone [140].

#### 6.2.6. IFN-α2b + Pulsed Dye Laser for Periungual Warts

A randomized study by Choi et al. (2002) demonstrated that combining intralesional IFN α-2b with pulsed dye laser (PDL) significantly improved treatment outcomes for recalcitrant periungual warts. The clearance rate was: 92.3% with combination therapy, 50% with IFN α-2b alone, 0% with PDL alone. No significant side effects were reported, highlighting the efficacy and safety of this synergistic approach [141].

#### 6.2.7. IFN-α2b + *Candida*/Mumps/*Trichophyton* Antigens

This randomized, single-blinded, placebo-controlled trial by Horn et al. (2005) evaluated the effectiveness of intralesional immunotherapy using skin test antigens (*Candida*, mumps, or *Trichophyton*) in 233 patients with warts, with or without IFN α-2b. The study showed that intralesional immunotherapy with *Candida*, mumps, or *Trichophyton* antigens is highly effective in the treatment of warts; however, the addition of IFN α-2b does not provide any additional therapeutic benefit [97].

#### 6.2.8. PPD + MMR + *Candida* Antigen

Triple intralesional immunotherapy by Nofal et al. (2022) using a combination of PPD, *Candida* antigen, and MMR vaccine showed superior efficacy in treating multiple recalcitrant warts compared to each agent alone. In a randomized study of 160 patients, 77.5% of those receiving the combined therapy achieved complete clearance, with the lowest recurrence rate, though not statistically significant. The treatment was found to be both safe and more effective, suggesting a promising option for resistant HPV-induced warts [142].

### 6.3. Intralesional HPV Vaccination in Recalcitrant Wart Immunotherapy

The intralesional administration of prophylactic HPV vaccines is emerging as a novel and promising strategy for the treatment of recalcitrant warts, offering a new immunotherapeutic avenue beyond their conventional preventive use. In clinical cases, a routine systemic administration of the 9-valent HPV vaccine has been reported to lead to the complete cure of all existing warts [143,144]. Another study showed that intralesional administration is more effective than systemic administration, possibly due to localized dendritic cell activation and the triggering of delayed-type hypersensitivity responses [140].

Unlike conventional intralesional therapies such as *Candida* antigen, MMR, or tuberculin, HPV vaccines exhibit potent immunogenic effects when injected directly into warts. Study assessing intralesional administration of the 9-valent HPV vaccine reported complete recalcitrant wart clearance in 60% of patients and significant improvement in the remaining 40%, with no recurrences observed during follow-up [144]. 

In a randomized controlled trial, Nofal et al. reported a 90% clearance rate with the quadrivalent HPV vaccine, significantly outperforming the bivalent formulation (30%) and saline control (0%) [145]. Supporting this, a 2024 randomized controlled trial found a 75% complete clearance rate using intralesional quadrivalent HPV vaccine in patients with recalcitrant non-genital warts, exceeding the efficacy of *Candida* antigen (40%) and placebo—see Table 9 [146]. The study by Elyamany et al. (2025) evaluated the efficacy of intralesional acyclovir versus the quadrivalent HPV vaccine in treating recalcitrant cutaneous warts. The study found that the intralesional quadrivalent HPV vaccine was significantly more effective and well-tolerated, establishing it as a superior immunotherapeutic option compared to acyclovir [147].

These therapeutic effects may be mediated by cross-reactive immune responses due to L1 capsid protein homology between vaccine-targeted and cutaneous HPV types, enhanced by the vaccine’s adjuvants. Additionally, the distant clearance of untreated warts suggests a systemic cell-mediated immune activation—particularly a Th1-skewed response involving IL-2 and IFN-γ [17,21,148]. Overall, while more robust, controlled studies are needed, these findings support the repurposing of HPV vaccines as therapeutic agents for recalcitrant warts, potentially in combination with other immunomodulatory treatments.

### 6.4. Clinical Trials

#### 6.4.1. Efficacy of Cryotherapy Combined with Intralesional Hepatitis B Virus Vaccine

Recent clinical trials have explored novel combinations and immunomodulatory strategies in the intralesional treatment of HPV-induced warts, aiming to overcome the limitations of traditional destructive methods. One promising approach is the integration of cryotherapy (representing a first line of therapy for cutaneous warts) with intralesional Hepatitis B virus (HBV) vaccine, evaluated in a comparative study assessing their combined versus separate efficacy in treating multiple cutaneous warts (NCT05902624). Although previous studies have demonstrated modest success rates (around 20–23%) with low-dose (0.2 mL) intralesional HBV vaccine, recent findings suggest that higher doses administered intramuscularly (0.5–1 mL) may significantly improve clearance rates (up to 50%), raising important questions regarding the optimal route and dosage for immunomodulation [149]. The HBV vaccine’s capacity to stimulate Th1-mediated immune responses, coupled with its excellent safety profile even in immunocompromised patients, positions it as a promising agent for further exploration. 

#### 6.4.2. Sclerotherapy and *Candida* Antigen in Treatment of Common Warts

Another clinical trial explores a novel, minimally invasive approach for treating common warts by comparing three intralesional therapies: *Candida* antigen, polidocanol sclerotherapy, and a combination of both. The trial is based on the idea that combining two mechanisms—immune stimulation and vascular destruction—could produce synergistic effects that improve wart clearance and reduce recurrence. The *Candida* antigen works by stimulating the body’s immune system, not just at the site of injection but also at distant wart locations, leading to a broader therapeutic effect [42]. Meanwhile, polidocanol—used as a sclerosing agent—targets the wart’s blood supply, causing vessel closure and depriving the lesion of nutrients, which leads to its gradual shrinkage and resolution. Conventional treatments like cryotherapy or electrocautery often involve pain, tissue damage, and higher recurrence. In contrast, this combined approach is designed to be less painful, more effective, and suitable for stubborn or widespread warts, potentially offering a safer, cost-effective alternative grounded in both immunological and vascular treatment principles (NCT06749665).

## 7. Conclusions

Cutaneous warts are one of the most common dermatological conditions, affecting a substantial part of the population, especially patients in younger age groups. Recent advances in immunotherapy have offered promising treatment options for patients who have not achieved satisfactory results with previous conventional modalities such as cryotherapy and salicylic acid. Emerging immunotherapeutic agents aim to modulate and exert an immune response, thereby promoting the clearance of HPV lesions. Intralesional immunotherapy works primarily through a type IV delayed-type hypersensitivity, initiating a complex interplay of cytokine-mediated responses. 

However, there are many limitations regarding the new therapeutic approach. Notably, there is a necessity to conduct randomized, double-blinded clinical trials and long-term follow-ups to assess the recurrence rate of treated warts and the efficacy of the method. Many existing studies involve a small sample size, which limits the general applicability of the findings. Furthermore, the absence of standardized proctoceles complicates evaluation and comparison across different modalities. 

Although generally considered highly effective, modalities such as *Candida* antigen, MMR, or PPD have been researched in many different studies; however, the results remain inconsistent. This variability emphasizes the need for further research to elucidate the factors contributing to different outcomes across studies and to identify the most effective and well-tolerated intervention. Despite current challenges, intralesional immunotherapy remains a promising and potentially transformative strategy for treating non-genital warts.

## Figures and Tables

**Figure 1 ijms-26-05644-f001:**
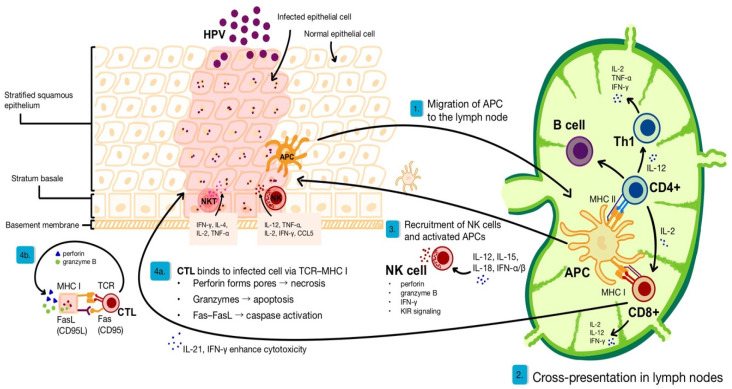
Immune-mediated clearance of HPV: interplay between innate and adaptive responses [18,19,20,21]. Abbreviations: HPV—human papillomavirus; APC—antigen-presenting cell; CTL—cytotoxic T lymphocyte; Th1—T helper type 1 cell; MHC I—major histocompatibility complex class I; MHC II—major histocompatibility complex class II; TCR—T cell receptor; FasL (CD95L)—Fas ligand; Fas (CD95)—Fas receptor; NK cell—natural killer cell; KIR—killer-cell immunoglobulin-like Receptor; IL—interleukin; IFN—interferon, TNF-α—tumour necrosis factor alpha.

**Figure 2 ijms-26-05644-f002:**
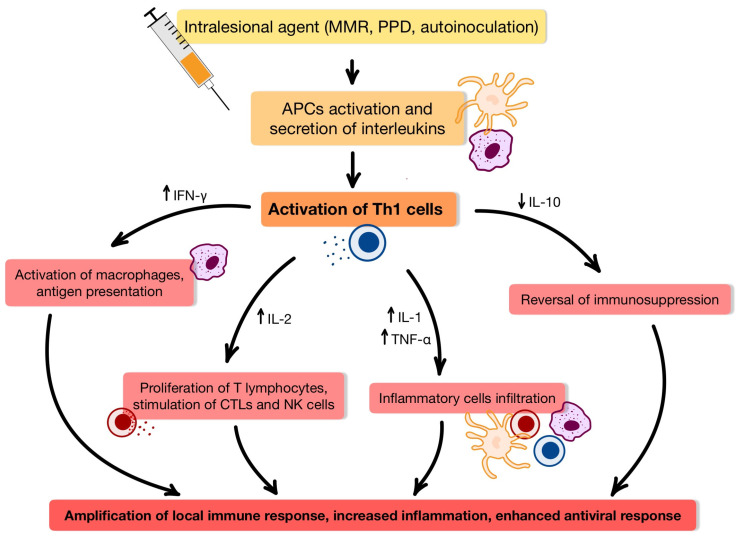
Mechanism of delayed-type IVa immune response in HPV wart clearance [37,41]. Macrophages and dendritic cells, which are activated antigen-presenting cells, secrete IL-12 and IL-18, which prime Th1 cells. Th1 cells then produce IFN gamma (IFN-γ), a central cytokine that activates macrophages and enhances their microbicidal function. Simultaneously, IL-2 supports the proliferation of T cells and boosts cytotoxic T lymphocyte (CTL) and natural killer (NK) cell activity, while TNF-α and IL-1 contribute to the recruitment of inflammatory cells and amplify local immune responses. In parallel, levels of IL-10—a regulatory cytokine that normally suppresses immune activation—are reduced, relieving immunosuppressive pressure and enabling a more robust clearance of HPV-infected cells. Importantly, although the immune stimulus is administered locally, the resulting activation is often systemic, allowing for the clearance of both treated and distant, uninjected lesions. This systemic effect offers a distinct advantage over conventional destructive treatments, which are typically confined to localized lesions and do not prevent recurrence. Abbreviations: MMR—Measles–Mumps–Rubella vaccine; PPD—Purified Protein Derivative; APCs—antigen presenting cells; IFN-γ—interferon gamma; TNF-α—tumor necrosis factor α; IL—interleukin; Th1—T helper cell type 1; CTLs—cytotoxic T lymphocyte; ↑—increase; ↓—decrease.

**Figure 3 ijms-26-05644-f003:**
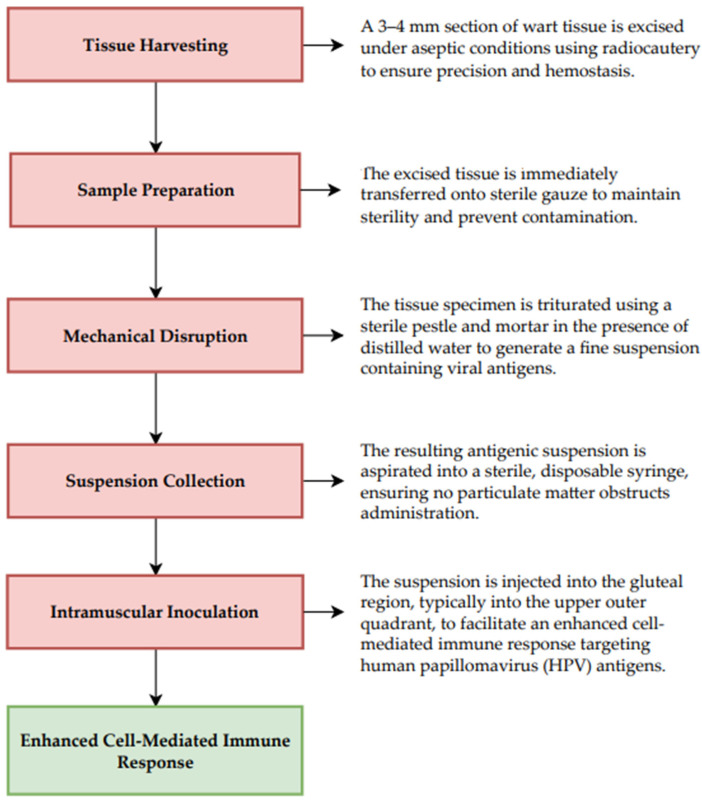
Protocol for Autoinoculation in the Management of Non-genital Viral Warts [74,75,76].

**Table 1 ijms-26-05644-t001:** HPV clearance mechanisms induced by interferons (IFN) [23,32,33].

IFN Type	Produced by	Key Immune Functions	HPV Evasion Strategy
IFN-α/β [23,32,33]	Keratinocytes, DCs	Induce IFN-stimulated genes (ISGs) that inhibit viral replication Enhance NK cell cytotoxicity Promote cross-priming of CD8^+^ T cells via dendritic cells	E6 degrades IRF3, a key transcription factor for IFN-β production E7 blocks ISGF3 complex, impairing ISG expression E5 inhibits STING pathway, further limiting IFN signalling
IFN-γ [32,33]	NK cells, NKT cells, activated CD4^+^ and CD8^+^ T cells	Enhances antigen presentation via upregulation of MHC class I and II Promotes Th1 immune polarization Activates macrophages and sustains CTL function	HPV reduces MHC I and II expression, diminishing the effects of IFN-γ Chronic low-level HPV presence can desensitize cells to IFN-γ signalling
IFN-λ [32,33]	Epithelial cells, DCs at mucosal/skin surfaces	Provide localized antiviral defence at epithelial barriers Less systemic inflammation compared to IFN-α/β Reinforce type I IFN responses and limit epithelial viral spread	HPV may downregulate IFN-λ receptors on keratinocytes HPV likely suppresses this pathway as well via E6/E7 interference with downstream JAK/STAT signalling

**Table 2 ijms-26-05644-t002:** Summary of Published Protocols for Intralesional MMR Vaccine Administration in the Treatment of Non-genital Viral Warts [50,51,52].

Author (Year)	Type of Warts	Injection Protocol	Follow-Up	Complete Clearance Rate
Shaker ESE (2021) [51]	Single or multiple cutaneous	Up to 3, every 3 weeks	6 months	30%
Kaur A (2021) [50]	Cutaneous	3 sessions, every 3 weeks	Every 4 weeks for 24 weeks	76.67%
Lahoria U. (2023) [52]	Extragenital cutaneous	Every 2 weeks, up to 7 or until clearance	Not specified	58%

**Table 3 ijms-26-05644-t003:** Cytokine modulation following intralesional PPD immunotherapy [57,58,59,60].

Study	Cytokines Affected	Direction of Change Post-PPD
Abd-Elazeim et al. (2014) [57]	IL-12	↑
Shaheen et al. (2015) [58]	IL-12	↑
Abou-Taleb et al. (2019) [59]	IFN-γ, IL-12	↑
Sil et al. (2021) [60]	IL-10	↓

IL—interleukin; IFN-γ—interferon gamma; Th1/Th2—T helper cell type 1/type 2; ↑—increase; ↓—decrease.

**Table 4 ijms-26-05644-t004:** Summary of clinical studies (2020–2025) investigating intralesional PPD, BCG, MWV immunotherapy for the treatment of non-genital warts [67,68,69,70,71,72,73].

Author (Year)	Type of Warts	Agent & Dose	Injection Protocol	Complete Clearance Rate
Fawzy et al. (2020) [67]	Plane	PPD 0.1 mL	Every 2 weeks, max 5 sessions	55%
Ghaly et al. (2021) [68]	Plantar	PPD 0.1 mL	Every 2 weeks, max 3 sessions	50%
Nofal et al. (2021) [69]	Periungual	PPD 0.1 mL	Every 2 weeks, max 5 sessions	70%
Rutnin et al. (2023) [70]	Palmoplantar & periungual	PPD 0.3 mL (20 IU/mL)	Every 2 weeks, max 5 sessions	80%
Ebrahim et al. (2021) [71]	Multiple	BCG 0.1 mL	Up to 5 sessions, 3 weeks apart	63.8%
Eldahshan et al. (2022) [72]	Recalcitrant extragenital	BCG 0.1 mL	Every 2 weeks, up to 5 sessions	70%
Dhakar et al. (2016) [73]	Refractory extragenital	Mw 0.1 mL	Into largest/3 lesions; weekly, max 12 or until clearance	66.7%

**Table 5 ijms-26-05644-t005:** Summary of clinical studies investigating autoinoculation for the treatment of non-genital cutaneous warts [39,40,75,76].

Author (Year)	Type of Warts	Injection Protocol	Follow-Up	Clearance Rate
Taneja et al. (2020) [78]	Multiple viral	3 sessions (months 0, 1, 2); fewer if cleared	Final check at 3 months	67%
Abdelmonaem et al. (2021) [79]	Multiple recalcitrant	1 session	Weeks 1, 2, 12, 16; based on size/number	66%
Ashraf et al. (2023) [40]	Multiple	Up to 3, at 1-month intervals	3 months after last session	91.66%
Shahid et al. (2023) [39]	Multiple viral	Max 3, monthly	Months 1, 2, 3; based on count reduction	77.1%

**Table 6 ijms-26-05644-t006:** Summary of clinical studies (2020–2025) investigating intralesional *Candida* antigen immunotherapy for the treatment of non-genital warts [55,67,88,89,90,91,92,93,94].

First Author (Year)	Type of Warts	*Candida* Solution Dose Per Session	Injection Protocol	Complete Clearance Rate
Fawzy MM (2020) [67]	Plane warts	0.3 mL of 1/100 solution	Every 2 weeks, max 5 sessions	76.7%
Marei A (2020) [88]	Recalcitrant warts	0.2 mL of 1/1000 solution	Every 2 weeks, 5 sessions	40%
Rageh RM (2021) [55]	Plantar warts	0.3 mL of 1:100 solution	Every 3 weeks, max 5 sessions	80%
Abdelaal MA (2021) [89]	Plantar warts	0.1 mL into largest wart	Every 3 weeks, up to 3 sessions	40%
Hodeib AAE (2021) [94]	Plane warts	0.3 mL	Every 2 weeks, up to 4 sessions	60%
Nofal A (2021) [90]	Common & plantar warts	0.2 mL into largest wart	Every 2 weeks, max 5 sessions	73.5%
Nofal A (2022) [91]	Plane warts	0.1 mL of 1/1000 solution	Every 2 weeks, max 5 sessions	55.6%
Nasr M (2023) [92]	Multiple warts	Not specified	Every 2 weeks, up to 5 sessions	60%
Youssef EMK (2023) [93]	Common, plantar, plane	0.1 mL of 1/100 and 1/1000 solutions	Not clearly defined	94.3% (1/100); 77.1% (1/1000)

**Table 7 ijms-26-05644-t007:** Summary of clinical studies (2020–2025) investigating intralesional vitamin D3 for the treatment of non-genital warts [52,92,102,103,105,106,107,108].

First Author (Year)	Type of Warts	Vitamin D3 Dose Per Session	Injection Protocol	Clearance Rate
Ibrahim NA et al. (2020) [105]	Recalcitrant palmoplantar	0.2 mL (300,000 IU/mL)	Up to 5 warts/session, monthly, max 4 sessions	88.89%
Zainab Z et al. (2021) [106]	Cutaneous	0.2 mL (15 mg/mL)	Every 2 weeks, 4 sessions	57.9%
Abdel Razik LH et al. (2021) [107]	Recalcitrant multiple common	0.6 mL (60,000 IU/wart)	Every 3 weeks until clearance	20%
Lahoria U et al. (2023) [52]	Extragenital cutaneous	0.2 mL (IU not specified)	Every 2 weeks, response monitored	64%
Nasr et al. (2023) [92]	Multiple	0.3 mL (100,000 IU/mL)	Every 2 weeks, up to 5 sessions	48%
Prathibha et al. (2023) [103]	Palmoplantar	0.2–0.5 mL (~15 mg/mL)	Every 2 weeks, up to 4 sessions	88.5%
Al-Sabak et al. (2023) [102]	Cutaneous	0.2 mL (600,000 IU/ampoule)	4 sessions every 2 weeks	81.9%
Almuhyi et al. (2024) [108]	Cutaneous	0.2–0.3 mL (300,000 IU/ampoule)	4 sessions every 2 weeks	59%

**Table 8 ijms-26-05644-t008:** Summary of Clinical Trials Published Between 2020 and 2025 on Intralesional Immunotherapy for Non-Genital Warts [50,55,67,70,92,107].

First Author (Year)	Compared Interventions	Categories of Treated Cutaneous Warts	Dosage	Regimen	Study Population	Complete Clearence Rate	Additional Remarks
Abdel Razik L, (2021) [107]	*Candida* antigen and vitamin D3	Recalcitrant multiple common warts	Group I: 0.3 mL of 1/100 *Candida* antigen solution Group II: 0.6 mL of cholecalciferol aqueous solution Group III: 0.3 mL of normal saline.	Solution injected into largest wart at 3-week intervals until complete clearance or for maximum 4 treatment sessions	80 patients with 30 patients assigned to group I and group II and 20 to group III	76.7% in *Candida* antigen group; 20% in vitamin D3 group	
Fawzy MM, (2020) [67]	PPD and *Candida* antigen and MMR vaccine	Plane warts	Group I: 0.1 mL of PPD Group II: 0.1 mL of 1/1000 solution of *Candida* antigen Group III: 0.1 mL of MMR vaccine	Patients were injected directly, without pre- sensitization. Injections were administered at 2-week intervals until complete clearance or for a maximum of 5 treatment sessions	120 patients, with 40 patients assigned to each treatment group	55% in the PPD group; 70% in the *Candida* antigen group 62.5% in the MMR group	Recurrence of warts was observed in 3 patients of the PPD group, while *Candida* antigen and MMR groups showed no recurrence of the lesions after the 6 month follow-up period
Kaur A, 2021 [50]	MIP and MMR vaccine	Cutaneous warts	Group A: 0.1 mL of MIP vaccine Group B: 0.5 mL of MMR vaccine	Injections without pre- sensitization. were repeated at intervals of 3 weeks, or until complete clearance or maximum of 3 treatment sessions	60 patients, with 30 patients assigned to each treatment group	90% in the MIP group; 76.67% in the MMR group	MIP showed an early response compared to MMR (9.41 vs. 11.71 weeks)
Nasr M, (2023) [92]	*Candida* antigen and vitamin D3	Multiple warts	Group A: 0.3 mL of 1/100 solution of *Candida* antigen Group B: 0.3 mL of cholecalciferol aqueous solution	Solution of *Candida* antigen was injected into the base of largest wart; cholecalciferol aqueous was injected into the base of each wart with maximum of 5 warts; Injections were administered at 2-weeks intervals until full clearance, or for a total of five sessions	50 patients with 25 patients assigned to each treatment group	60% in the *Candida* antigen group; 48% in vitamin D3 group	*Candida* antigen is superior in common warts, palmoplantar warts and subungual warts
Rageh RM, (2021) [55]	*Candida albicans* antigen and MMR vaccine	Plantar warts	Group A: 0.3 mL of 1/100 solution of *Candida* antigen Group B: 0.3 mL of MMR vaccine	Solution injected into largest wart at 3-week intervals until complete clearance or for maximum 5 treatment sessions	60 patients, with 30 patients assigned to each treatment group	80% in the *Candida* antigen group; 26.7% in the MMR group	41.7% patients with concomitant distant warts in Group A and 12.5% in Group B showed resolution of their distant un-injected warts
Rutnin S, (2023) [70]	MMR vaccine and PPD	Palmoplantar and Periungual Warts	Group A: 0.3 mL of MMR vaccine Group B: 0.3 mL of PPD 20 iu/mL	Injection to the wart or largest wart in patients with multiple lesions; reinjection every 2 weeks until complete clearance or a maximum of 5 injections	40 patients, with 20 patients assigned to each treatment group	90% in the MMR treated group; 80% in PPD treated group	MMR showing trend toward faster clearance

**Table 9 ijms-26-05644-t009:** Clinical and distant response rates to intralesional therapies for recalcitrant non-genital warts—Fouda et al., 2024 [146].

Treatment Group	Complete Response	Partial Response	No Response	Distant Response (Complete)	Distant Response (Partial)
HPV Vaccine	75%	0%	25%	72.7%	27.3%
*Candida* Antigen	40%	15%	45%	33.3%	50%
Placebo (Saline)	5%	15%	80%	11.1%	22.2%

## Data Availability

Not applicable.

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
