# Peer review of "Intralesional Immunotherapy for Non-Genital Viral Warts: A Review of Current Evidence and Future Perspectives"

_ijms, 2025, doi:10.3390/ijms26125644_

Round 1

Reviewer 1 Report

Comments and Suggestions for Authors

The review article by Kucharczyk et al was on the whole well written and reviewed the use of various intralesional immunotherapies to treat non genital warts.  This should be a review that is likely to generate much interest if published. The background to the topic was well written.

There were a few issues that need to be addressed in order to bring it up to publication standard. In particular a close attention to the cited references and their appearance in the reference list.  Many references do not line up against the cited reference number.  Also some references do nto align to the stated scientific finding. 

Comments for attention

Line 180-188  The Dock8 deficiency can also impact Th cell differentiation (decreased Th1/Th17  and increased Th2) and B cell responses to  (disrupt GC responses by interfering with BCR signalling) which are also important to note  as humoral  immunity will be important to control HPV infections as well.

Line 235.  The reference 41 cited does not mention anything about a type IV DTH response and so may need to select a different reference to support this statement.

Line 238 – reference 6 doesn’t mention anything about MMR vaccine may need to choose a more appropriate reference.

Line 248 delete the word protocol

Line 376.  The section 3.4.2 needs to provide a brief overall summary of the finding so f the autoinoculation with wart antigens

Lin 405- need to define what is meant by BK viremia

Section 3.5  Is there any information to explain why Candida antigens induce donor specific Abs? Is this in anyway linked to the wart antigens as well?

Section 3.6  Need to provide a little bit of back to Trichophyton and how common human infections are and what is known about Trichophyton antigens which are immunodominant in humans and are these used in the treatment modality described in this section?

Line 456- The reference Vance at al is not listed at reference 120 in the reference list.

Line 464 is citing a secondary reference- need to cite the primary research article associated with the study involving IFNa injections

Line 585  The sentence should rad :  In a trial….

Lin 671   The reference cited is incorrect - it should be cited as 131

Line 768  incorrect citation for Nofal et al

Table 9 Incorrect citation for Foudat et al

Author Response

Dear Reviewer,

We would like to sincerely thank you for your thorough and constructive review of our manuscript. We are grateful for your positive assessment of the overall structure and clarity of our work, as well as your recognition of the clinical relevance and potential interest this review may generate.

We also appreciate your careful reading and the detailed comments regarding reference alignment, scientific accuracy, and areas requiring further clarification or expansion.

We are currently addressing each of the points you raised and will submit a revised version along with a detailed point-by-point response shortly:

Line 180-188  The Dock8 deficiency can also impact Th cell differentiation (decreased Th1/Th17  and increased Th2) and B cell responses to  (disrupt GC responses by interfering with BCR signalling) which are also important to note  as humoral  immunity will be important to control HPV infections as well.

We have introduced an additional paragraph elaborating on the immunological consequences of Dock8 deficiency, including decreased Th1/Th17 to Th2 balance and impaired germinal center (GC) responses due to altered BCR signaling.

Line 235.  The reference 41 cited does not mention anything about a type IV DTH response and so may need to select a different reference to support this statement.

            We have implemented the attached changes in the references.

Line 238 – reference 6 doesn’t mention anything about MMR vaccine may need to choose a more appropriate reference.

            We have implemented the attached changes in the references

Line 248 delete the word protocol

            The changes have been implemented in accordance with the recommendation.

Line 376.  The section 3.4.2 needs to provide a brief overall summary of the finding so f the autoinoculation with wart antigens

            The changes have been implemented in accordance with the recommendation.

Lin 405- need to define what is meant by BK viremia

            The changes have been implemented in accordance with the recommendation.

Section 3.5  Is there any information to explain why Candida antigens induce donor specific Abs? Is this in anyway linked to the wart antigens as well?

            The study cited in our manuscript reported this phenomenon but did not explore or explain the underlying immunological mechanism responsible for it. To our knowledge, this is the only documented case in the current literature addressing such an outcome in this specific context. As such, the connection between Candida antigens and donor-specific antibody induction remains unclear and warrants further investigation.

Section 3.6  Need to provide a little bit of back to Trichophyton and how common human infections are and what is known about Trichophyton antigens which are immunodominant in humans and are these used in the treatment modality described in this section?

            We have followed your recommendation and incorporated a new paragraph providing an overview of the prevalence of Trichophyton infections in humans, the known immunodominant antigens, and their relevance to the therapeutic approach discussed in this section. We believe this addition enhances the clarity and context of the described treatment modality.

Line 456- The reference Vance at al is not listed at reference 120 in the reference list.

            The changes have been implemented in accordance with the recommendation.

Line 464 is citing a secondary reference- need to cite the primary research article associated with the study involving IFNa injections

            The changes have been implemented in accordance with the recommendation.

Line 585  The sentence should rad :  In a trial….

            The changes have been implemented in accordance with the recommendation.

Lin 671   The reference cited is incorrect - it should be cited as 131

            The changes have been implemented in accordance with the recommendation.

Line 768  incorrect citation for Nofal et al

            The changes have been implemented in accordance with the recommendation.

Table 9 Incorrect citation for Foudat et al

            The changes have been implemented in accordance with the recommendation.

Thank you once again for your time, expertise, and thoughtful feedback.

With kind regards,

Authors

Reviewer 2 Report

Comments and Suggestions for Authors

Dear Authors, 

I was pleased to read this interesting review. Genital or non-genital warts are a common problem in dermatology practice and of great interest due to the lack of an effective treatment.

The authors managed to structure very well the available therapeutic options at this moment and to make a synthesis of the specialized literature. Although the effectiveness of intralesional immunotherapy has been proven superior for several years now, the dynamics in this field compel us to stay updated with the novelties and add value to the article.

The paper is well structured, contains relevant figures and tables for the subject, and the bibliographic references comply with the requirements.

I consider the approach an original one and I suggest publishing in its current form.

Author Response

Dear Reviewer,

We would like to express our deepest gratitude for your thoughtful review of our manuscript.

Your recognition of the clinical significance and therapeutic complexity surrounding both genital and non-genital warts is greatly appreciated. We are truly honored by your kind remarks regarding the clarity, structure, and scientific rigor of our work. It was our intention to present a comprehensive and current synthesis of available treatment modalities, and your acknowledgement affirms the value of that effort.

With sincere appreciation,

Authors